# Pneumonia remains a leading public health problem among under-five children in peri-urban areas of north-eastern Ethiopia

**Awoke Keleb**[1]☯, **Tadesse Sisay**[1], **Kassahun Alemu**[2], **Ayechew Ademas**[1], **Mistir Lingerew**[1], **Helmut Kloos**[3], **Tefera Chane Mekonnen**[4], **Atimen Derso**[1], **Metadel Adane**[1]☯*

1 Department of Environmental Health, College of Medicine and Health Sciences, Wollo University, Dessie, Ethiopia, 2 Department of Epidemiology and Biostatistics, Institute of Public Health, University of Gondar, Gondar, Ethiopia, 3 Department of Epidemiology and Biostatistics, University of California, San Francisco, CL, United States of America, 4 Department of Nutrition, School of Public Health, College of Medicine and Health Sciences, Wollo University, Dessie, Ethiopia

☯ These authors contributed equally to this work.
* metadel.adane2@gmail.com

**Data Availability Statement:** All relevant data are within the manuscript and its Supporting Information files.

## Abstract

### Background

Pneumonia is a leading cause of morbidity and mortality among children under five years of age in developing countries, including Ethiopia. However, data on this serious illness among highly susceptible and vulnerable children living in local peri-urban areas are limited. Establishing the prevalence of pneumonia and identifying the associated factors are important for proper planning and intervention.

### Methods

A community-based cross-sectional study was conducted among 560 systematically selected children under the age of five years in peri-urban areas of Dessie City from January through March 2019. Data were collected using a pretested structured questionnaire, physical examination of children and direct observation of housing conditions. Pneumonia was examined using World Health Organization (WHO) guidelines as the presence of the symptoms of fast breathing or indrawn chest with or without fast breathing during the two weeks prior to the study. A principal component analysis was used to construct a household wealth index. Data were analyzed using a binary logistic regression model at 95%CI (confidence interval). The analysis involved estimating the crude odds ratio (COR) using bivariate analysis, and adjusted odds ratio (AOR) using multivariable analysis. From the multivariable analysis, variables at *p*-value of less than 0.05 were declared statistically significant.

### Main findings

The prevalence of pneumonia among children under five was 17.1% (95%CI: 13.9%-19.9%). Of the participating children, 113 (21.0%) had a cough, 92 (17.1%) had fast breathing, 76 (14.1%) had fever, and 40 (7.4%) of the children had chest indrawn. Domestic fuel was the most common source of cooking fuel 383 (71.1%). Majority 445 (82.6%) of children

**Funding:** This study was funded by Wollo University. The funders had no role in study design, data collection and analysis, decision to publish, or preparation of the manuscript.

**Competing interests:** The authors have declared that no competing interests exist.

**Abbreviations:** COR, crude odds ratio; AOR, adjusted odds ratio; ARI, acute respiratory infection; CI, confidence interval; EDHS, Ethiopia Demographic and Health Survey; SD, standard deviations; IQR, Interquartile range; WHO, World Health Organization.

were fully vaccinated and 94 (17.4%) were not fully vaccinated. Most (481, 89.2%) of the children were got exclusive breastfeeding. Slightly more than half (284, 52.7%) of the under-five children had acute malnutrition and 27.1% of the children had a childhood history of ARI. The multivariable analysis showed using domestic fuel as the energy source for cooking (adjusted odds ratio [AOR] = 3.95, 95%CI: 1.47–10.62), cooking in the living room (AOR = 6.23; 95%CI: 1.80–21.68), overcrowding (AOR = 3.37, 95%CI: 1.56–7.27), child history of acute respiratory infection (ARI) (AOR = 6.12 95%CI: 2.77–13.53), family history of ARI (AOR = 4.69, 95%CI: 1.67–13.12) and acute malnutrition (AOR = 2.43, 95%CI: 1.18–5.04) were significantly associated with childhood pneumonia.

## Conclusion

In this study, pneumonia remains a leading public health problem among under five children in the study area and higher than national averages. Domestic fuel as the energy source for cooking, cooking in the living room, overcrowding, child history of ARI, family history of ARI and acute malnutrition were predictors of pneumonia. Community-based interventions focusing on improving housing conditions, reduced use of domestic biofuels, adequate and balanced food intake, including exclusive breastfeeding of infants, and early treatment of ARIs.

## Introduction

Pneumonia is a severe form of acute lower respiratory infection that is responsible for high morbidity and mortality rates among children under five; it poses a major threat to public health worldwide [1, 2]. Globally, pneumonia is the leading cause of child mortality, responsible for approximately 6.0% of the 5.9 million deaths in the under-five age group, killing around 900,000 children in 2015. It accounts for the loss of over 2,500 children's lives every day, or over 100 every hour [3].

Pneumonia is highly prevalent in sub-Saharan Africa and South Asia; 50.0% of total deaths from pneumonia worldwide in 2015 occurred in six countries: India, Nigeria, Pakistan, Democratic Republic of Congo (DRC), Ethiopia, and China [4]. More than 490,000 children under five died of pneumonia in 2016 in sub-Saharan Africa [3, 5]. Despite interventions and sustained efforts from a range of stakeholders in Ethiopia, pneumonia is still a leading single cause of under-five morbidity and mortality, constituting 18.0% of all causes of mortality and killing over 40,000 children in this age group in the country every year [4]. Amhara (where Dessie is located) is one of the most affected region, with pneumonia accounting for 8.0% of all acute respiratory infections (ARIs) among children under five; the national prevalence is 7.0% [6].

Previous studies have identified several common factors associated with pneumonia in children under-five, including overcrowding, several other environmental conditions, malnutrition and poverty, absence of ventilation, and indoor air pollution [7, 8]. However, the influence of these factors varies among different populations of children. For example, indoor air pollution tends to be worse in peri-urban communities where biomass fuels are more frequently used in cooking and heating due to lack of access to other forms of energy [9].

Peri-urbanization affects hundreds of millions of children worldwide who are being raised in overcrowded and unhygienic conditions and poor housing structures that characterize

many peri-urban areas. Such conditions facilitate the transmission of pneumonia and other diseases, driving up child mortality [10, 11]. Evidence on the prevalence and determinants of pneumonia in children under five in peri-urban areas of Dessie City is scarce. Furthermore, children are more likely to develop pneumonia than other age groups. Therefore, the aim of this study was to assess the prevalence and associated factors of pneumonia among children under five in peri-urban areas of Dessie City Administration in north-eastern Ethiopia.

## Methods and materials

### Study design, period and setting

A community-based cross-sectional study design was conducted in six peri-urban *kebeles* of Dessie City Administration from January through March 2019. Dessie is located about 400 km northeast of Ethiopia's capital city of Addis Ababa in Amhara Regional State. Dessie City Administration has 16 *kebeles*, 10 urban and 6 peri-urban *kebeles* (*kebele* is the smallest administrative unit in Ethiopia, each with around 5,000 people). Based on the national census, Dessie City Administration had a total population of 212,436 in 2014. Of the total, 34,748 (16.4%) lived in peri-urban *kebeles* [12]. Dessie City is located at an elevation above 2,400 meter above sea level. Most of the total area of the city is mountainous (60%) and 40% is a plateau [13].

The health facilities in Dessie City's urban *kebeles* include 1 government referral hospital, 1 government general hospital, 3 private hospitals, 8 health centers, 33 private clinics, 5 wholesale pharmacies, 18 public pharmacies and 33 private pharmacies. The peri-urban *kebeles* have only 2 rural drug venders and 6 health posts [14].

### Source population, and inclusion and exclusion criteria

The source populations for this study consists of mother/caregiver-child of under five years of age living in the six peri-urban *kebeles* of Dessie City. Children under five with mother/caregiver who were lived in the six peri-urban *kebeles* of Dessie City during the two weeks prior to data collection were included. However, those under-five children with mother/caregiver who were away during the two weeks prior to the survey but available during the data collection day were excluded since they may acquire pneumonia in that place where they stayed.

### Sample size determination and sampling methods

The sample size was determined using the single population proportion formula [15] considering the assumptions of:

$$n = \frac{(z_{a/2})^2 * p(1-p)}{d^2}$$

$Z_{\alpha/2}$ is the standard normal variable value at (1-α) % confidence level (α is 0.05 with 95%CI [confidence interval], $Z_{\alpha/2}$ = 1.96), *p* is an estimate of the prevalence of pneumonia (33.3%), which is taken from a similar study conducted in Wondo Genet Town, southern Ethiopia [16] and *d* margin of error (4.0%). Adjusting for an anticipated 5.0% non-response rate, the final sample size was determined to be 560.

The 560 study participants were proportionally allocated for the six *kebeles* based on the number of children under five in each *kebele*. Systematic random sampling with an interval of 6 was used to select households in each *kebele*. The first household was selected using the lottery method. For those households with more than one child under five, one child was selected randomly. Households in which study participants were not available at the first visit were

revisited once more the same day or the following day. If a participant was still not available, he/she was considered as a non-respondent.

## Outcome and explanatory variables

The outcome variable was the presence or absence of pneumonia in a child under five. Explanatory variables were socio-economic and demographic variables, housing and environmental variables, nutritional and immunization variables and pre-existing medical conditions.

## Operational definitions

**Peri-urban area.** Partly urbanized area of population located consolidated urban and rural regions [11, 17].

**Pneumonia.** Defined based on World Health Organization (WHO) classifications, children with fast breathing or indrawn chest with or without fast breathing during the two weeks prior to the study were classified as having pneumonia, whereas children with cough and colds who did not have fast breathing and had no indrawn chest during the two weeks prior to the study were classified as having no pneumonia [18].

**Fast breathing.** Defined as 60 or more breaths per minute for children less than 2 months old, 50 breaths or more per minute for those 2–12 months old, and 40 breaths or more per minute for those 12–59 months old [18].

**Fever.** Defined as elevated axillary body temperature of 37.5˚C or above [19].

**Family history of Acute Respiratory Infection (ARI).** A household with a history of pneumonia, bronchitis, ear infection, common cold, tonsillitis, or pharyngitis confirmed by a clinician during the 15 days prior to data collection.

**Full or incomplete vaccination.** **Full vaccination** includes all children who had obtained BCG (bacillus calmette–guérin vaccine) and OPV0 (oral polio vaccine) at birth, pentavalent 1 (DPT-hepB-Hib [diphtheria, pertussis, tetanus, hepatitis B and haemophilus influenzae type b]), OPV1, and PCV1 at 6 weeks; pentavalent 2, OPV2, PCV2 at 10 weeks; pentavalent 3, PCV3 at 14 weeks; and measles vaccine at 9 months; **incomplete vaccination** includes those children without up-to-date or only partial vaccination [6, 20].

**Overcrowding.** A house having an area per person of less than 75 square feet [17].

## Data collection, management and quality assurance

A pretest and structured questionnaire used for this study by modifying the British Medical Research Council's (BMRC) questionnaire for pneumonia [21] after validated in the context of the local culture, language and others. The questionnaire was adopted and was in English version, translated to Amharic (local language), and translated back to English to ensure consistency. A pre-test was conducted using a 5.0% sample size of the total study sample in peri-urban *kebeles* of Kombolcha Town (near Dessie City) to establish the validity and reliability of the questionnaire. The questionnaire was amended based on the findings of the pre-test.

The data collectors were trained focused on the survey instrument, physical examination, and measuring the mid-/upper-arm circumference. The data collectors were six female public health officers and administered face-to-face interviews with mothers and/or primary caregivers; observation of participants' housing and environmental conditions and physical examination of the children.

First, mothers/caregivers were asked whether the child under observation had a cough with fast breathing and/or chest indrawn in the two weeks preceding the survey. Second, a physical examination was performed that entailed measuring axillary temperature and measuring respiratory rate using a timer. To confirm results, the respiratory rate count was repeated two

to three times for each child and the count without disturbance was taken. Finally, nutritional status was determined by measuring mid-/upper-arm circumference using a standard measuring tape. The measurement was taken twice and the average value to the nearest 0.1cm was recorded. Supervision was performed for data quality control.

The completeness and consistency of the questionnaires was checked daily during data collection. Then, data were entered using EpiData version 3.1 and 10.0% of the questionnaires were randomly re-checked to identify data entry errors. Once the data entry was completed, the data was exported to the Statistical Package of the Social Science (SPSS) version 25.0 for data cleaning and analysis. Basic data quality assurance measures were taken, including data cleaning using browsing of data tables after sorting, graphical exploration of distributions using box plots, histograms, and scatter plots, frequency distributions and cross tabulations, summary statistics and statistical outlier detection using sorting were performed. Descriptive statistics were used for categorical variables and mean ±SD (standard deviations) and/or median (IQR, interquartile range) for continuous variables. Continuous variables were categorized using information from the literature and categorical variables were re-categorized accordingly.

## Statistical analysis

A principal component analysis was used to construct a household wealth index (low, medium and high categories) after checking its assumptions for communality value $> 0.5$, KMO (sampling adequacy) $> 0.5$, which was 0.757 with $p$-value $<0.001$ and complex structure factor (eigenvalue) greater than 1. Bivariate (crude odds ratio [COR]) and multivariable (adjusted odds ratio [AOR]) values were calculated using logistic regression analysis with 95% confidence interval [CI]. From the bivariate analysis, variables with $p < 0.25$ were considered for multivariable analysis.

From the multivariable logistic regression analysis, variables with a significance level of $p < 0.05$ were taken as statistically significant and independently associated with under-five pneumonia. The presence of multi-collinearity among independent variables was checked using standard error at the cutoff value of 2, and we found a maximum standard error of 1.51, indicated no multi-collinearity. None of the covariates were collinear (Pearson's correlation coefficient $r > 0.7$). Model fitness was checked using the Hosmer-Lemeshow test, which had a $p$-value 0.823.

## Ethical consideration

The ethical approval letter was obtained from the Institutional Ethical Review Committee of the College of Medicine and Health Sciences of Wollo University. An informed consent was obtained from mothers/caregivers. Children who were sick during data collection were linked to the nearest health facility for further treatment. They were assured that their information would not be used for purposes other than scientific research and the participation was voluntary and that they could withdraw from the interview at any time for whatever reason. Confidentiality was maintained by avoiding possible identifiers.

## Results

### Socio-demographic and economic characteristics of participants

Of the 560 under-five children, 539 participated in the study (96.25% response rate). The wealth index 180 (33.4%) of the study participants were under economically medium, whereas

179 (33.2%) within low category. Nearly half 245 (45.5%) of the mothers/caregivers had primary education. One-fourth 136 (25.2%) of the households had five or more persons.

The mean age of mothers/caregivers was 29.56 ± 5.56 years and the median age of children aged 0–59 months was 20.0 (15.3 [IQR]) months, with 54.5% of the children younger than 24 months (Table 1).

### Housing and environmental characteristics

Of the 539 respondents, the majority 437 (81.1%) lived-in privately-owned houses. Domestic fuel (wood, charcoal or kerosene) 383 (71.1%) was the most common source of cooking fuel, and 289 (53.62%) of children were carried by mothers/caregivers during cooking. About 479 (88.9%) households had a separate room used as the kitchen; only 163 (34.0%) of these had a window. A total of 60 (11.1%) households used their living room for cooking. Other housing and environmental characteristics and the results of the bivariate analysis of their association with childhood pneumonia are summarized in Table 2.

### Immunization and nutritional characteristics of participants

Of the 539 participating children, 445 (82.6%) were fully vaccinated and 94 (17.4%) were not fully vaccinated. Most 481 (89.2%) of their mothers/caregivers practiced exclusive breastfeeding in the first six months of the child's life. Almost two-thirds (330, 61.2%) of the children received zinc supplementation and 841 (89.2%) of them received vitamin A supplements (Table 3).

### Pre-existing medical conditions

Nearly half ($n$ = 255, 47.3%) of the children had normal nutritional status and 284 (52.7%) had acute malnutrition. ARI was the most common medical condition; 146 participants (27.1%) had a childhood history of ARI and 41 (7.6%) had a family history of ARI. The great majority of the children had no history of TB (tuberculosis) 525 (97.4%) and asthma 515 (95.5%) (Table 4).

### Prevalence, and signs and symptoms of pneumonia

The overall prevalence of pneumonia was 17.1% (95%CI: 13.9%-19.9%). Of the participating children, 113 (21.0%) had a cough, 92 (17.1%) had fast breathing, 76 (14.1%) had fever, and 40 (7.4%) of the children had chest indrawn (Fig 1).

### Factors associated with pneumonia among children under five

After adjusting for confounding variables from multivariable logistic regression analysis, our results indicate that under-five children whose families cooked their food in the living room were 6.23 times more likely to develop childhood pneumonia (AOR = 6.23; 95%CI: 1.80–21.68) compared to than participants whose families cooked their food in the kitchen. The odds of pneumonia in children under five was 3.95 times (AOR = 3.95; 95%CI: 1.47–10.62) higher for participants whose families used domestic fuel such as wood, charcoal, and kerosene for cooking than those that used electricity (Table 5).

The odds of pneumonia among children under five living in overcrowded conditions were 3.37 times higher than among participants not living in overcrowded spaces (AOR = 3.37; 95% CI: 1.56–7.27). Children with a history of ARI were 6.12 times more likely to develop pneumonia (AOR = 6.12; 95%CI: 2.77–13.53), and the odds for children with a family history of ARIs were 4.69 times (AOR = 4.69; 95%CI: 1.67–13.12) higher than for their counterparts without a

**Table 1. Bivariate analysis of the association of socio-demographic factors with pneumonia among children under five in peri-urban areas of Dessie City, north-eastern Ethiopia, January—March 2019.**

| Variable | Frequency | Pneumonia | | COR (95% CI) | P-value |
|---|---|---|---|---|---|
| | | Yes | No | | |
| | n (%) | n | n | | |
| **Age of mother/caregiver (years)*** | | | | | |
| 15–19 | 22 (4.1) | 8 | 14 | 2.41(0.90–6.42) | 0.079 |
| 20–34 | 397 (73.6) | 61 | 336 | 0.77(0.45–1.30) | 0.324 |
| 35–49 | 120 (22.3) | 23 | 97 | Ref | |
| **Age of child (months)¥** | | | | | |
| 0–11 | 152 (28.2) | 24 | 128 | 0.87(0.39–1.90) | 0.726 |
| 12–23 | 142 (26.3) | 24 | 118 | 0.94(0.43–2.07) | 0.884 |
| 24–35 | 99 (18.4) | 12 | 87 | 0.64(0.26–1.55) | 0.324 |
| 36–47 | 84 (15.6) | 21 | 63 | 1.55(0.68–3.50) | 0.297 |
| 48–59 | 62 (11.5) | 11 | 51 | Ref | |
| **Sex of child** | | | | | |
| Female | 268 (49.7) | 55 | 213 | 1.63(1.04–2.58) | 0.035 |
| Male | 271 (50.3) | 37 | 234 | Ref | |
| **Religion** | | | | | |
| Christian | 42 (7.8) | 14 | 28 | 2.69(1.35–5.33) | 0.005 |
| Muslim | 497 (92.2) | 78 | 419 | Ref | |
| **Birth order of child** | | | | | |
| First | 168 (31.2) | 25 | 143 | Ref | |
| Second | 168 (31.2) | 30 | 138 | 1.24(0.70–2.22) | 0.462 |
| Third | 122 (22.6) | 21 | 101 | 1.18(0.63–2.24) | 0.592 0.333 |
| Fourth or above | 81(15.0) | 16 | 65 | 1.41(0.70–2.81) | |
| **Mother's/caregiver's marital status** | | | | | |
| Unmarried | 46 (8.5) | 15 | 31 | 2.61(1.35–5.07) | 0.004 |
| Married | 493 (91.5) | 77 | 416 | Ref | |
| **Mother's/caregiver's educational status** | | | | | |
| Cannot read and write | 91(16.9) | 20 | 71 | 0.93(0.35–2.47) | 0.877 |
| Read and write | 39(7.2) | 9 | 30 | 0.97(0.32–3.04) | 0 .980 |
| Primary level | 245(45.5) | 40 | 205 | 0.64(0.26–1.59) | 0.339 |
| Secondary level | 134(24.9) | 16 | 118 | 0.45(0.16–1.20) | 0.111 |
| Diploma or above | 30(5.6) | 7 | 23 | Ref | |
| **Mother's/caregiver's occupational status** | | | | | |
| Housewife | 346(64.2) | 47 | 299 | Ref | |
| Civil servant | 35(6.5) | 12 | 23 | 3.32(1.55–7.12) | 0.002 |
| Day laborer | 50(9.3) | 14 | 36 | 2.47(1.24–4.93) | 0.010 |
| Merchant | 57(10.6) | 9 | 48 | 1.19(0.55–2.59) | 0.656 |
| Other | 51(9.5) | 10 | 41 | 1.55(0.73–3.31) | 0.255 |
| **Father's educational status** | | | | | |
| Cannot read and write | 74 (15.0) | 13 | 61 | 0.87(0.36–2.08) | 0.754 |
| Read and write | 30 (6.1) | 11 | 19 | 2.36(0.89–6.26) | 0.084 |
| Primary level | 161 (32.7) | 20 | 141 | 0.58(0.26–1.27) | 0.173 |
| Secondary level | 167 (33.9) | 21 | 146 | 0.59(0.27–1.28) | 0.181 |
| Diploma or above | 61 (12.4) | 12 | 49 | Ref | |
| **Father's occupational status** | | | | | |
| Unemployed | 42 (8.5) | 8 | 34 | 1.13(0.43–2.95) | 0.809 |

*(Continued)*

**Table 1.** (Continued)

| Variable | Frequency | Pneumonia | | COR (95% CI) | *P*-value |
|---|---|---|---|---|---|
| | | Yes | No | | |
| | *n* (%) | *n* | *n* | | |
| Farmer | 179 (36.3) | 18 | 161 | 0.54(0.25–1.14) | 0.104 |
| Day laborer | 116 (23.5) | 24 | 92 | 1.25(0.60–2.59) | 0.552 |
| Merchant | 75 (15.2) | 14 | 62 | 1.01(0.44–2.30) | 0.994 |
| Civil servant | 81 (16.4) | 14 | 67 | Ref | |
| **Household size (persons)** | | | | | |
| >5 | 136 (25.2) | 26 | 110 | 1.21(0.73–1.99) | 0.463 |
| ≤5 | 403 (74.8) | 66 | 337 | Ref | |
| **Economic wealth index** | | | | | |
| Low | 179 (33.2) | 25 | 154 | 0.49(0.28–0.84) | 0.009 |
| Medium | 180 (33.4) | 22 | 158 | 0.42(0.24–0.73) | 0.002 |
| High | 180 (33.4) | 45 | 135 | Ref | |

Ref, Reference category.

*Mean age of mothers/caregivers was 29.56 ± 5.56 [SD] years.

¥Median age of children aged 0–59 months was 20.0 (15.3 [IQR]) months.

family history of ARI. The odds of developing childhood pneumonia were 2.43 times higher in children with acute malnutrition than in children with normal nutritional status (AOR = 2.43; 95%CI: 1.18–5.04) (Table 5).

## Discussion

In this community-based cross-sectional study, we investigated the prevalence of pneumonia and its associated factors among children under five in peri-urban areas of Dessie City. We found the prevalence of pneumonia to be 17.1%. The study also revealed using domestic fuel as the energy source for cooking, cooking in the living room, overcrowding, child history of ARI, family history of ARI and acute malnutrition were factors significantly associated with pneumonia among under-five children in peri-urban areas of Dessie City.

This rate of pneumonia in our study was almost twice as high as the prevalence of ARI (7.0%) among similar children reported by the 2016 Ethiopia Demographic and Health Survey (EDHS) [6]. This discrepancy might be because the national prevalence statistic is an aggregate of different acute upper and lower respiratory infections. However, this rate was lower than the prevalence of pneumonia reported by studies conducted in Jimma Zone (28.1%) [22] and Wondo Genet District (33.3%) in Ethiopia [16]. These differences might be due to differences in study settings, environmental factors, the basic infrastructure of study households, and socio-demographic characteristics of mothers/caregivers. Furthermore, the higher prevalence of pneumonia in our study than in the 2016 EDHS might have been due to the repeated history of ARI, overcrowding and higher levels of indoor air pollution from greater use of domestic biofuels.

The prevalence in this study is similar to the pneumonia prevalence (16.1%) reported by a community-based cross-sectional study conducted in Este Town of South Gondar Zone, Amhara Region, Ethiopia [23]. Our finding is also consistent with that of a community-based cross-sectional study in Dibrugarh Town, India (16.34%) [24]. These similarities may reflect similarities in study settings. In this study, a total of 492 children received the pneumonia vaccine at their recommended time; 75 (15.2%) of these children developed pneumonia. This may

**Table 2. Bivariate analysis of the associations of housing and environmental factors with pneumonia among children under five in peri-urban areas of Dessie City, north-eastern Ethiopia, January—March 2019.**

| Variable | Frequency | Pneumonia | | COR (95% CI) | P-value |
| --- | --- | --- | --- | --- | --- |
| | | Yes | No | | |
| | n (%) | n | n | | |
| **House ownership** | | | | | |
| Rent | 102 (18.9) | 37 | 65 | 3.95(2.42–6.47) | <0.001 |
| Private ownership | 437 (81.1) | 55 | 382 | Ref | |
| **Type of walls in home** | | | | | |
| Wood with mud | 412 (76.4) | 80 | 332 | 2.31(1.21–4.39) | 0.011 |
| Stone and mud cement/bricks | 127 (23.6) | 12 | 115 | Ref | |
| **Type of floor in home** | | | | | |
| Earth | 354 (65.7) | 73 | 281 | 2.27(1.32–3.89) | 0.003 |
| Cement/ceramic | 185 (34.3) | 19 | 166 | Ref | |
| **Does kitchen have a window** | | | | | |
| No | 316 (66.0) | 49 | 267 | 2.12(1.11–4.03) | 0.022 |
| Yes | 163 (34.0) | 13 | 150 | Ref | |
| **Number of windows per house** | | | | | |
| ≤2 windows | 273 (50.6) | 71 | 202 | 4.10(2.43–6.91) | <0.001 |
| >2 windows | 266 (49.4) | 21 | 245 | Ref | |
| **Separate room used for kitchen** | | | | | |
| No | 60 (11.1) | 30 | 30 | 6.73(3.79–11.92) | <0.001 |
| Yes | 479 (88.9) | 62 | 417 | Ref | |
| **Place of cooking** | | | | | |
| Living room | 75 (13.9) | 38 | 37 | 7.79(4.57–13.30) | <0.001 |
| Kitchen | 464 (86.1) | 54 | 410 | Ref | |
| **Type of fuel used for cooking** | | | | | |
| Domestic fuel (wood, charcoal/kerosene) | 383 (71.1) | 81 | 302 | 3.54(1.83–6.84) | <0.001 |
| Electricity | 156 (28.9) | 11 | 145 | Ref | |
| **Location of the child during cooking** | | | | | |
| Carried on mother's/caregiver's back | 289 (53.6) | 71 | 218 | 3.55(2.11–5.98) | <0.001 |
| Out of the cooking area | 250 (46.4) | 21 | 229 | Ref | |
| **Family cigarette smoking** | | | | | |
| Yes | 74 (13.7) | 26 | 48 | 3.28(1.90–5.64) | <0.001 |
| No | 465 (86.3) | 66 | 399 | Ref | |
| **Number of persons per room** | | | | | |
| >2 person per room | 112 (20.8) | 42 | 70 | 4.52(2.79–7.33) | <0.001 |
| ≤2 person per room | 427 (79.2) | 50 | 377 | Ref | |
| **Overcrowding status** | | | | | |
| Overcrowded | 231 (42.9) | 65 | 166 | 4.08(2.50–6.64) | <0.001 |
| Not overcrowded | 308 (57.1) | 27 | 281 | Ref | |

Ref, Reference category.

be due to the vaccine losing its potency because of poor vaccine stock management and poor vaccine handling.

Our multivariable logistic model indicated that cooking in the living room, using domestic fuel for cooking, overcrowding, a child's history of ARI, a family history of ARI and acute malnutrition were significantly associated with pneumonia among children under five. Children living in households using the living room for cooking were more likely to have pneumonia

**Table 3. Bivariate analysis of the association of nutritional and immunization factors with pneumonia among children under five in peri-urban areas of Dessie City, north-eastern Ethiopia, January—March 2019.**

| Variable | Frequency | Pneumonia | | COR (95% CI) | P-value |
|---|---|---|---|---|---|
| | | Yes | No | | |
| | n (%) | n | n | | |
| **Vitamin A supplementation** | | | | | |
| No | 58 (10.8) | 23 | 35 | 3.92(2.18–7.04) | <0.001 |
| Yes | 481 (89.2) | 69 | 412 | Ref | |
| **Zinc supplementation** | | | | | |
| No | 209 (38.8) | 51 | 158 | 2.28(1.44–3.58) | <0.001 |
| Yes | 330 (61.2) | 41 | 289 | Ref | |
| **Pneumonia vaccination** | | | | | |
| No | 47 (8.7) | 17 | 30 | 3.15(1.65–5.99) | <0.001 |
| Yes | 492 (91.3) | 75 | 417 | Ref | |
| **Vaccination status of child** | | | | | |
| Incomplete | 94 (17.4) | 30 | 64 | 2.89(1.74–4.82) | <0.001 |
| Fully vaccinated | 445 (82.6) | 62 | 383 | Ref | |
| **Breastfeeding** | | | | | |
| Mixed feeding | 58 (10.8) | 29 | 29 | 6.64(3.72–11.84) | <0.001 |
| Exclusive feeding | 481 (89.2) | 63 | 418 | Ref | |
| **Total months of breastfeeding** | | | | | |
| <12 months | 192 (35.6) | 37 | 155 | 1.27(0.80–2.01) | 0.313 |
| ≥12 months | 347 (64.4) | 55 | 292 | Ref | |

Ref, Reference category.

than those households using a separate kitchen. This finding is consistent with results from worldwide systematic reviews and meta-analyses [2]. It suggests that cooking in the living room imposes a high level of indoor air pollution and suffocation that could increase the incidence of pneumonia among children under five. Therefore, cooking in a separate kitchen appears to be important for improving child survival.

In our findings, children living in households that used domestic fuel for cooking had higher odds of developing pneumonia than children from households that used electricity for cooking. This finding is supported by studies in Este Town in northwestern Ethiopia [23], Wolayta-Sodo in southern Ethiopia [25], and northeast Brazil [26]. The use of traditional cooking fuels increases indoor air pollution, and since children under five spend most of their time with their mothers/caregivers as they cook, the children are exposed to biomass fuel pollution, which increases the incidence of pneumonia.

In this study, household overcrowding was found to be a predictor of childhood pneumonia. This result supports findings from studies from northeast Brazil [27], Canada [28], and India [29, 30] that reported higher pneumonia rates in children living in overcrowded conditions. A systematic review revealed that household crowding has a uniform risk worldwide, with odds ratios between 1.9 and 2.3 in the low- middle- and high-income countries [27]; these ratio are lower than our findings (AOR = 3.37). The difference might be due to poorer housing conditions, smaller living spaces, and a larger number of families per household in our study.

In this study, a child's history of ARI and a family history of ARI both predicted higher odds of a child having pneumonia compared to the absence of a child's or family's ARI. This

**Table 4. Bivariate analysis of the association of pre-existing medical conditions with pneumonia among children under five in peri-urban areas of Dessie City, north-eastern Ethiopia, January—March 2019.**

| Variables | Frequency | Pneumonia | | COR (95% CI) | *P*-value |
|---|---|---|---|---|---|
| | | Yes | No | | |
| | *n* (%) | *n* | *n* | | |
| **Child history of ARI** | | | | | |
| Yes | 146 (27.1) | 46 | 100 | 3.47(2.18–5.52) | <0.001 |
| No | 393 (72.9) | 46 | 347 | Ref | |
| **Family history of ARI** | | | | | |
| Yes | 41 (7.6) | 22 | 19 | 7.08(3.65–13.75) | <0.001 |
| No | 498 (92.4) | 70 | 428 | Ref | |
| **Child history of CHD** | | | | | |
| Yes | 11 (2.0) | 5 | 6 | 4.22(1.26–14.15) | 0.019 |
| No | 528 (98.0) | 87 | 441 | Ref | |
| **Child history of HIV/AIDS** | | | | | |
| Yes | 14 (2.6) | 7 | 7 | 5.18(1.77–15.14) | 0.003 |
| No | 525 (97.4) | 85 | 440 | Ref | |
| **Child history of TB** | | | | | |
| Yes | 14 (2.6) | 8 | 6 | 7.00(2.37–20.69) | <0.001 |
| No | 525 (97.4) | 84 | 441 | Ref | |
| **Child history of asthma** | | | | | |
| Yes | 24 (4.5) | 12 | 12 | 5.44(2.36–12.53) | <0.001 |
| No | 515 (95.5) | 80 | 435 | Ref | |
| **Nutritional status of child** | | | | | |
| Acute malnutrition | 284 (52.7) | 70 | 214 | 3.46(2.07–5.79) | <0.001 |
| Normal | 255 (47.3) | 22 | 233 | | |

1, Reference category; CHD, Congenital heart disease; ARI, Acute respiratory infection; HIV, Human immunodeficiency virus.

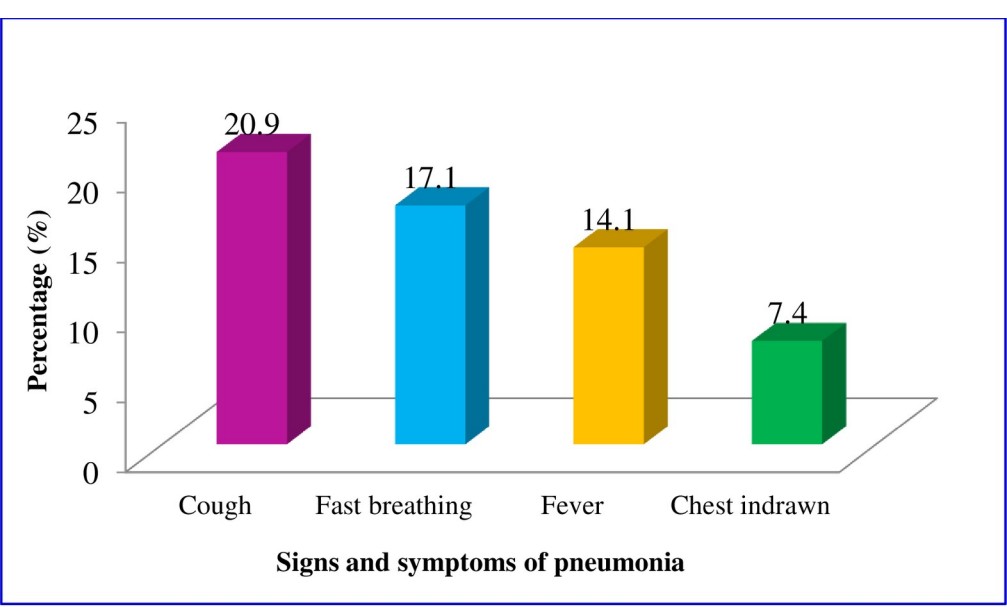

**Fig 1. Signs and symptoms of childhood pneumonia in peri-urban areas of Dessie City, north-eastern Ethiopia, January—March 2019.**

**Table 5. Factors significantly associated with pneumonia among children under five from multivariable logistic regression analysis in peri-urban areas of Dessie City, north-eastern Ethiopia, January—March 2019.**

| Variable | Pneumonia | | COR (95% CI) | AOR (95% CI) |
|---|---|---|---|---|
| | Yes | No | | |
| | n | *n* | | |
| **Place of cooking** | | | | |
| Living room | 38 | 37 | 7.79(4.57–13.30) | 6.23(1.80–21.68) |
| Kitchen | 54 | 410 | Ref | Ref |
| **Type of fuel used for cooking** | | | | |
| Domestic fuel (wood, charcoal/kerosene) | 81 | 302 | 3.54(1.83–6.84) | 3.95(1.47–10.62) |
| Electricity | 11 | 145 | Ref | Ref |
| **Overcrowding status** | | | | |
| Overcrowded | 65 | 166 | 4.08(2.50–6.64) | 3.37(1.56–7.27) |
| Not overcrowded | 27 | 281 | Ref | Ref |
| **Child history of ARI** | | | | |
| Yes | 46 | 100 | 3.47(2.18–5.52) | 6.12(2.77–13.53) |
| No | 46 | 347 | Ref | Ref |
| **Family history of ARI** | | | | |
| Yes | 22 | 19 | 7.08(3.65–13.75) | 4.69(1.67–13.12) |
| No | 70 | 428 | Ref | Ref |
| **Nutritional status of child** | | | | |
| Acute malnutrition | 70 | 214 | 3.46(2.07–5.79) | 2.43(1.18–5.04) |
| Normal | 22 | 233 | Ref | Ref |

Ref, Reference category.

*Variables adjusted for the multivariable analysis were: Age of mother/caregiver (years); child age (months); child gender; religion; mother's/caregiver's education and occupation; father's education and occupation; economic status (wealth index); house ownership; wall material; floor material; number of windows per house; cooking location; type of fuel used; location of child during cooking; family cigarette smoking; persons per room; overcrowding status; vitamin A supplementation; zinc supplementation; child PCV; vaccination status; breastfeeding for 6 months; parent history of ARI; child history of ARI, CHD (Congenital heart disease), HIV, TB and asthma, and nutritional status.

finding is consistent with those of studies conducted in Oromia Zone, Ethiopia [31]; Kenya [32]; and India [33]. In all these studies, children who had concomitant infections may have had their immunity lowered, making them more susceptible to pneumonia; ARIs are very contagious and easily transmitted.

Our data revealed that acute malnutrition carried higher odds of pneumonia than good nutritional status. This result is concordant with earlier studies conducted in Kersa District, southwest Ethiopia [34]; Pakistan [35]; and India [29, 33]. A systematic reviews and meta-analyses study findings also revealed that children with malnutrition are more likely to develop pneumonia than children with normal nutritional status [27, 36]. The similar findings across studies might be due to similar low food security levels and inadequate feeding practices in the study areas.

## Limitation of the study

This study was not used chest radiography and blood culture and/or culture of bronchi alveolar lavage to confirm pneumonia so that this study may not as strong as pneumonia confirmation using laboratory diagnostic tools. However; to overcome such limitation, we diagnosed pneumonia based on the 2014 standard clinical WHO and integrated management of new-

born child illness classification of cases [18]. We also operationalized variables, used appropriate protocols and trained professional data collectors to assure data quality.

Our study may over report some behaviors due to social desirability bias during self-reporting. However, we tried to control social desirability bias by employing proxy data collectors because proxy subjects may yield reliable information about behavior of target persons. Furthermore, the level of accuracy of the measuring instrument was revised after pre-testing the questionnaire data collection tools. The findings of this study may not be representative of the peri-urban areas at the national level as the study was conducted only in peri-urban areas of Dessie City and establishing a temporal relationship between the risk factors and the outcome was also impossible.

Longitudinal studies covering different seasons may provide a better understanding of the occurrence of pneumonia in peri-urban areas of Dessie City and help to guide interventions. Further studies based on chest radiography and blood cultures and/or cultures of bronchi alveolar lavage to confirm the presence of pneumonia are highly encouraged.

## Conclusion

This study found the prevalence of pneumonia in peri-urban areas of Dessie City to be higher than that of ARIs nationally and pneumonia to be a common disease among children under five in the study area. Cooking in the living room, using domestic fuel for cooking, overcrowding, a child's history of ARI, a family's history of ARI, and acute malnutrition were found to be factors significantly associated with pneumonia.

These findings provide strong evidence that pneumonia can be prevented through community-based interventions that achieve ventilated and improved housing conditions, separate kitchens, less use of domestic biofuels, adequate and balanced food intake, including exclusive breastfeeding of infants, and early treatment of ARIs. The high prevalence of pneumonia in our study might be a result of deficiencies of the community- based pneumonia care and prevention programs and failure to adequately cover peri-urban communities. We therefore recommend implementation of a comprehensive health care program at the community level in the study area.

## Supporting information

**S1 File. Household survey questionnaire in English version.**
(DOCX)

**S2 File. Household survey questionnaire in Amharic version.**
(DOCX)

## Acknowledgments

First and foremost, we thank Dessie City Administration Health Bureau and each peri-urban *kebele* administrator for allowing us to conduct the study and for providing information. We also appreciate and thank the data collectors and supervisors for their assistance and the mothers/caregivers for their cooperation during data collection.

## Author Contributions

**Conceptualization:** Awoke Keleb, Tadesse Sisay, Ayechew Ademas, Mistir Lingerew, Metadel Adane.

**Data curation:** Awoke Keleb, Metadel Adane.

**Formal analysis:** Awoke Keleb, Metadel Adane.

**Funding acquisition:** Awoke Keleb, Metadel Adane.

**Investigation:** Awoke Keleb, Tadesse Sisay, Metadel Adane.

**Methodology:** Awoke Keleb, Tadesse Sisay, Kassahun Alemu, Ayechew Ademas, Mistir Lingerew, Tefera Chane Mekonnen, Atimen Derso, Metadel Adane.

**Project administration:** Awoke Keleb, Metadel Adane.

**Resources:** Awoke Keleb, Tadesse Sisay, Kassahun Alemu, Ayechew Ademas, Mistir Lingerew, Helmut Kloos, Tefera Chane Mekonnen, Atimen Derso, Metadel Adane.

**Software:** Awoke Keleb, Tadesse Sisay, Kassahun Alemu, Mistir Lingerew, Tefera Chane Mekonnen, Atimen Derso, Metadel Adane.

**Supervision:** Awoke Keleb, Tadesse Sisay, Kassahun Alemu, Metadel Adane.

**Validation:** Awoke Keleb, Tadesse Sisay, Kassahun Alemu, Ayechew Ademas, Mistir Lingerew, Helmut Kloos, Tefera Chane Mekonnen, Atimen Derso, Metadel Adane.

**Visualization:** Awoke Keleb, Tadesse Sisay, Kassahun Alemu, Ayechew Ademas, Mistir Lingerew, Helmut Kloos, Atimen Derso, Metadel Adane.

**Writing – original draft:** Awoke Keleb, Metadel Adane.

**Writing – review & editing:** Awoke Keleb, Helmut Kloos, Tefera Chane Mekonnen, Metadel Adane.

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
