## [Decision Letter · Decision Letter 0]

20 Mar 2020

PONE-D-19-33652

Prevalence and associated factors of pneumonia among children under five in peri-urban areas of Dessie City, northeastern Ethiopia: A community-based cross-sectional study

PLOS ONE

Dear Dr Adane (PhD),

Thank you for submitting your manuscript to PLOS ONE. After careful consideration, we feel that it has merit but does not fully meet PLOS ONE’s publication criteria as it currently stands. Therefore, we invite you to submit a revised version of the manuscript that addresses the reviewers' points raised below during the review process.

We would appreciate receiving your revised manuscript by May 04 2020 11:59PM. To enhance the reproducibility of your results, we recommend that if applicable you deposit your laboratory protocols in protocols.io, where a protocol can be assigned its own identifier (DOI) such that it can be cited independently in the future. For instructions see: http://journals.plos.org/plosone/s/submission-guidelines#loc-laboratory-protocols

We look forward to receiving your revised manuscript.

Kind regards,

Ray Borrow, Ph.D., FRCPath

Academic Editor

PLOS ONE

Journal Requirements:

2. Please address the following:

- Please include additional information regarding the survey or questionnaire used in the study and ensure that you have provided sufficient details that others could replicate the analyses. For instance, if you developed a questionnaire as part of this study and it is not under a copyright more restrictive than CC-BY, please include a copy, in both the original language and English, as Supporting Information.

- Please provide additional details regarding participant consent. In the ethics statement in the Methods and online submission information, please ensure that you have specified how verbal consent was documented and witnessed.

3. Please ensure that you refer to Figure 2 and 4 in your text as, if accepted, production will need this reference to link the reader to the figure.

Reviewers' comments:

Reviewer's Responses to Questions

**Comments to the Author**

1. Is the manuscript technically sound, and do the data support the conclusions?

Reviewer #1: Yes

Reviewer #2: Yes

Reviewer #3: Partly

2. Has the statistical analysis been performed appropriately and rigorously? 

Reviewer #1: Yes

Reviewer #2: Yes

Reviewer #3: I Don't Know

3. Have the authors made all data underlying the findings in their manuscript fully available?

Reviewer #1: Yes

Reviewer #2: Yes

Reviewer #3: Yes

4. Is the manuscript presented in an intelligible fashion and written in standard English?

Reviewer #1: Yes

Reviewer #2: Yes

Reviewer #3: Yes

5. Review Comments to the Author

Reviewer #1: Table 1 is missing.The serial starts from Table 2. However table 1 is referenced in the legend.To attach the table 1.

Justify how weaning period of 12 or more months is taken as reference category with WHO recommendations of weaning at 6 months of age.

Reviewer #2: The manuscript entitle ¨Prevalence and associated cactors of pneumonia among children under five in peri-urban areas of Dessie City, northeastern Ethiopia: A community-based cross sectional study¨ described by Kelleb A. et al., is very well written, interesting with values information about pneumonia. It is very difficult to perform a field study , moreover in Kebeles population. The authors accumulated an impressive data that should be used for others studies and diseases.

The questionnaire and analyses were intensely performed and the results and conclusions included all comments of the observational study.

In my opinion, the authors should develop more the subject about vaccination. It is a relevant thematic in the pneumonia domain and some information are missed, such us, the definition about fully vaccinated¨ and ¨not fully vaccinated. I am not aware about the vaccination system in Ethiopia.

Congratulations for this beautiful study.

Reviewer #3: The study is about children with clinical signs of pneumonia not confirmed cases of pneumonia since no microbological or radiological diagnosis done.

Table 1,2,3,4,5 and 6 tne first line and column 2 it is write (Pnrumonia N = 539) this is confusing, 539 is it the number of participant or number of children with pneumonia ? What are the meaning of yes and no of this colum ? What about presence and absence of pneumonia knowng that 92 children had pneumonia ?

Table 4 : the pneumonia vaccin and age at which it was administred was not mentionned in the study and out of 492 (75+417) children vaccinated against pneimonia 417 (yes) have pneumonia. Vaccination and occurence of pneumonia is an important point that should be discussed ?

The word Kebele have been used many times without giving the meaning.

Line 314-315 comparism of pneumonia prevalence of localised peri-urban area of Ethiopia to that of National level of Brazil done. What is the aim of this ?

6. PLOS authors have the option to publish the peer review history of their article (what does this mean?). If published, this will include your full peer review and any attached files.

Reviewer #1: Yes: Dr Gothankar J S

Reviewer #2: Yes: Glaucia Paranhos-Baccala

Reviewer #3: Yes: Ibrahim Dan Dano

---

## [Author Response · Author response to Decision Letter 0]

6 Jun 2020

Rebuttal letter

Response to the Journal Requirements Questions 

Question #1: Please ensure that your manuscript meets PLOS ONE's style requirements, including those for file naming.

Response: Thank you for this remark. We re-formatted the revised manuscript using the PLoS ONE format guidelines. The whole content of the manuscript, including the abstract, introduction, methods, discussion and reference are formatted using the guidelines (please see the revised version for each section).

Question #2. Please include additional information regarding the survey or questionnaire used in the study and ensure that you have provided sufficient details that others could replicate the analyses. For instance, if you developed a questionnaire as part of this study and it is not under a copyright more restrictive than CC-BY, please include a copy, in both the original language and English, as Supporting Information.

Response: We provided the questionnaire in original language and English version as supporting information S1 and S2. 

Question #3. - Please provide additional details regarding participant consent. In the ethics statement in the Methods and online submission information, please ensure that you have specified how verbal consent was documented and witnessed.

Response: We provided all the ethical consideration sub-heading (please see page 10 from lines 245 to 250. 

Question#4. Please ensure that you refer to Figure 2 and 4 in your text as, if accepted, production will need this reference to link the reader to the figure.

Response: We did the citation of Fig 1, 2 and 3 in the texts. One figure is reduced during the revision. Then the total figure becomes 3 not 4. 

Line by line response to reviewers

Reviewer # 1

Question #1. Table 1 is missing. The serial starts from Table 2. However table 1 is referenced in the legend. To attach the table 1. 

Response: Thank you for this comment and we included Table 1. 

Question #2. Justify how weaning period of 12 or more months is taken as reference category with WHO recommendations of weaning at 6 months of age.

Response: Sorry for the confusion we did. We studied about the total months for breast feeding as less than 12 months and 12 months and above. Then using the 12 months as a reference category for analysis (See Table 4). 

Reviewer #2

Reviewer #2: The manuscript entitle ¨Prevalence and associated cactors of pneumonia among children under five in peri-urban areas of Dessie City, northeastern Ethiopia: A community-based cross sectional study¨ described by Kelleb A. et al., is very well written, interesting with values information about pneumonia. It is very difficult to perform a field study, moreover in Kebeles population. The authors accumulated an impressive data that should be used for others studies and diseases. The questionnaire and analyses were intensely performed and the results and conclusions included all comments of the observational study. In my opinion, the authors should develop more the subject about vaccination. 

Question #1 It is a relevant thematic in the pneumonia domain and some information are missed, such us, the definition about fully vaccinated¨ and ¨not fully vaccinated. I am not aware about the vaccination system in Ethiopia. 

Response: Thank you for this pertinent comment. In Ethiopia, fully vaccination means the child received BCG and OPV0 at birth, pentavalent 1 (DPT-hib, he-b), OPV1, and PCV1 at 6 weeks, pentavalent 2, OPV2, PCV2 at 10 weeks, pentavalant 3, and PCV3 at 14th weeks and measles at 9 months, whereas incomplete vaccination includes those children in the status of up-to-date, partial and totally unvaccinated (See in page 8 from lines 186 to 190)

Reviewer #3

Reviewer #3: The study is about children with clinical signs of pneumonia not confirmed cases of pneumonia since no microbiological or radiological diagnosis done. 

Question #1. Table 1,2,3,4,5 and 6 the first line and column 2 it is write (Pnrumonia N = 539) this is confusing, 539 is it the number of participant or number of children with pneumonia ? What are the meaning of yes and no of this column? What about presence and absence of pneumonia knowing that 92 children had pneumonia? 

Response: 

Thank you for these important questions. We made a correction for the columns of each table and we deleted N = 539. However, Yes or No in the column is about to indicate the number of pneumonia and no pneumonia among children under-five. Absence of pneumonia means “No” and presence of pneumonia means “Yes”. Please see the revised Tables 1-5. 

Question # 2. Table 4 : the pneumonia vaccine and age at which it was administered was not mentioned in the study and out of 492 (75+417) children vaccinated against pneumonia 417 (yes) have pneumonia. Vaccination and occurrence of pneumonia is an important point that should be discussed? 

Response: In this study, a total of 492 children took the pneumonia vaccine at their respective age, but from 492 only 75 children were developing pneumonia. This may due to vaccines losing their potency because of poor vaccine stock management, poor vaccine handling and storage at storage centers even if they were potent on arrival (please see the revised version in page 14 from lines 349 to 352). 

Question # 3. The word Kebele have been used many times without giving the meaning. 

Response: We defined kebele and please see the revised version page 5 from lines 121 to 122.

Question #4. Line 314-315 comparison of pneumonia prevalence of localized peri-urban area of Ethiopia to that of National level of Brazil done. What is the aim of this?

Response: We thank for this important question, and we accept the comment and deleted the comparison since our study setting in a small area where the Brazil is national survey. 

We would like to thank the reviewers and editors for evaluating our manuscript. We have tried to address all the concerns in a proper way and believe that our paper has been improved considerably. We would be happy to make further corrections if necessary and look forward to hearing from you all soon. 

I hope that the revised manuscript is accepted for publication in PLoS ONE. 

Sincerely yours,

Metadel Adane (PhD)

---

## [Decision Letter · Decision Letter 1]

24 Jun 2020

Prevalence and associated factors of pneumonia among children under five in peri-urban areas of Dessie City, northeastern Ethiopia

PONE-D-19-33652R1

Dear Dr. Adane (PhD),

We’re pleased to inform you that your manuscript has been judged scientifically suitable for publication and will be formally accepted for publication once it meets all outstanding technical requirements.

Kind regards,

Ray Borrow, Ph.D., FRCPath

Academic Editor

PLOS ONE

Additional Editor Comments (optional):

Reviewers' comments:

Reviewer's Responses to Questions

**Comments to the Author**

1. If the authors have adequately addressed your comments raised in a previous round of review and you feel that this manuscript is now acceptable for publication, you may indicate that here to bypass the “Comments to the Author” section, enter your conflict of interest statement in the “Confidential to Editor” section, and submit your "Accept" recommendation.

Reviewer #1: All comments have been addressed

Reviewer #2: All comments have been addressed

Reviewer #3: All comments have been addressed

2. Is the manuscript technically sound, and do the data support the conclusions?

Reviewer #1: Yes

Reviewer #2: Yes

Reviewer #3: Yes

3. Has the statistical analysis been performed appropriately and rigorously? 

Reviewer #1: Yes

Reviewer #2: Yes

Reviewer #3: Yes

4. Have the authors made all data underlying the findings in their manuscript fully available?

Reviewer #1: Yes

Reviewer #2: Yes

Reviewer #3: Yes

5. Is the manuscript presented in an intelligible fashion and written in standard English?

Reviewer #1: Yes

Reviewer #2: Yes

Reviewer #3: Yes

6. Review Comments to the Author

Reviewer #1: (No Response)

Reviewer #2: The revised manuscript is much more clear after inclusion of all comments suggested by the reviewers. All questions were answered and the revised manuscript is now ready to be accepted and published.

Pneumonia studies in children under 5 years of age are primordial in peri-urbans regions. The intrinsic conditions of life were the children are living was described and pointed out as a major indicator for pneumonia risk factors. Others risk factors such as malnutrition despite are known, it is relevant to be compared with urban area. The study design and study management were performed very well. This kind of study is very complex and in this study, it was conducted very seriously. It is important to have this date in Ethiopia since no major data was described until now from Ethiopian researchers.

Thank you very much to give me the opportunity to review this manuscript.

Reviewer #3: All my comments have been taken into account and I am satisfied with the corrections done. The article is well written and provide valus informations about pneumonie.

7. PLOS authors have the option to publish the peer review history of their article (what does this mean?). If published, this will include your full peer review and any attached files.

Reviewer #1: **Yes: **Dr Jayashree Gothankar

Reviewer #2: **Yes: **Gláucia Paranhos-Baccalà

Reviewer #3: **Yes: **Dan Dano Ibrahim

---

## [Editor Report · Acceptance letter]

28 Jul 2020

PONE-D-19-33652R1 

Pneumonia remains leading public health problem among under-five children in peri-urban areas of northeastern Ethiopia 

Dear Dr. Adane (PhD):

I'm pleased to inform you that your manuscript has been deemed suitable for publication in PLOS ONE. Congratulations! Your manuscript is now with our production department. 

Kind regards, 

on behalf of

Prof. Ray Borrow 

Academic Editor

PLOS ONE